

**Geological Stratigraphy and Spatial Distribution of**
**Microfractures over Costa Rica Convergent Margin, Central**
**America- A Wavelet-Fractal Analysis**
Upendra K. Singh, Thinesh Kumar and Rahul Prajapati,
Department of Applied Geophysics, Indian School of Mines, Dhanbad-826 004, India
Correspondence: upendra_bhu1@rediffmail.com
**Abstract**
Identification of spatial variation of lithology, as a function of position and scale, is very
critical job for lithology modelling in industry. Wavelet Transform (WT) is an efficacious
and powerful mathematical tool for time (position) and frequency (scale) localization. It has
numerous advantages over Fourier Transform (FT) to obtain frequency and time information
of a signal. Initially Continuous Wavelet Transform (CWT) is applied on gamma ray logs of
two different Well sites (Well-1039 & Well-1043) of Costa Rica Convergent Margin, Central
America for identifications of lithofacies distribution and fracture zone later Discrete Wavelet
Transform (DWT) applied to DPHI log signals to show its efficiency in discriminating small
changes along the rock matrix irrespective of the instantaneous magnitude to represent the
fracture contribution from the total porosity recorded. Further the data of the appropriate
depths partitioned using above mathematical tools are utilized separately for WBFA. As
consequences of CWT operation it is found that there are four major sedimentary layers
terminated with a concordant igneous intrusion passing through both the wells. In addition of
WBFA analysis, it is clearly understanding that the fractal dimension value is persistent in
first sedimentary layers and the last gabbroic sill intrusions. Inconsistent value of fractal
dimension is attributed to fracture dominant in intermediate sedimentary layers it is also
validate through core analysis. Fractal Dimension values suggest that the sedimentary



environments persisting in that well locations bears abundant shale content and of low energy
environments.
**Key words**: CWT, DWT, Fractal, Costa Rica.

**1.  Introduction**
The nature of any log signal is fluctuating type in accordance to the subsurface geology. A
gamma ray log is most vividly used log for lithology identifications. These signals are very
noisy in some cases and highly fluctuating in another way. Manual interpretations of such
signals are quite difficult and it needs more experience. These difficulties are minimised by
kind of wavelet transform method. In our study Continuous Wavelet transform (CWT) is
tested on generated synthetic signals and applied to field data. The analysed results prove that
the CWT is highly suitable in geophysical log signals whereas conventional Fast Fourier
Transform fails in this case because it considers the whole signal in a stationary form.
Though Wavelet Transform provides unambiguous results in analysing the noisy and non-
stationary signals, its efficiency of extracting the information from the signal was seen
through its wavelet coefficients (Hui and Zaixing, 2010) with wavelet scalogram. Number of
publication has come to identify the lithofacies/boundary using various mother Wavelet
transform and Fourier transform (Chandrasekhar et al., 2012; Coconi et al., 2010; Dashtian,
2011; Javid and Tokhmechi, Mansinha et al.,1997; Mansinha 2003, 2004; Pan et al.,2007,
2012; Pinnegar and Stockwell, 2007; Stockwell et al., 1996; Sahimi and Hashemi, 2001;
Tokhmechi et al., 2009a, b; Yue et al., 2004; Zhang et al., 2011;).

In this paper, CWT and Discrete wavelet transform (DWT) are used separately for identifying
the lithology using gamma ray log data of well site 1039 and 1043 obtained from Costa Rica
Convergent Margin, Central America and computed wavelet scalograms. Moreover, the


information of fractures zones is analyzed with DWT using density logs data for both wells
that provides well featured whereas the log data doesn't carry information of fracture remains
featureless. Afterward, a linear relationship is obtained between the fracture density obtained
through DWT and identified fractures from water saturation logs using above methods. Apart
from wavelet analysis, one of the approach wavelet based fractal analysis techniques applied
to attribute the roughness/smoothness of the fractures. The obtained suggest that wavelet
transform acts as a microscope to delineate the high and low frequency hidden in the signal
separately, wavelet/holder exponent and fractal dimension are highly useful in identification
of lithofacies and spatial distribution of fractures.

**2.  Mathematical Background**
**2.1 Wavelet Transform**
Wavelet transform is mathematical tool that can be used to analyse both stationary and non-
stationary signals (Daubechies 1990, 1992) and expand time series into time frequency space.
Therefore, this method can find localized intermittent periodicities. For analysing stationary
or non-stationary signal proper mother wavelet has to be substituted and the operation of
continuous wavelet transform (CWT) proceeds as the convolution between time series of our
interest. The Discrete wavelet transform (DWT) is very useful in case of noisy data it
compresses the data by reducing noise and improve the resolution whereas the application of
CWT is preferring to extract the lithological feature from data. As it exposes the signal to
high and low frequency filters to form approximate and detailed coefficients traces out the
abrupt changes in the signal (Figure 8a and 8b). Basically, in geophysical well logs the abrupt
change corresponds to its own individual parameter changes which provide us more
information about the subsurface stratigraphy. This methodology pertaining to DWT allows
us to locate the high frequency changes immersed in the log which cannot be identified


manually. For example, gamma ray log is a good lithology indicator but in certain conditions
it is highly fluctuating in nature. This nature sometimes perturbs its evaluation. Apart from
lithology identification, DWT provides an advantage of analysing the fracture identifications.

**2.2 Continuous Wavelet Transform**
The concept of continuous wavelet transform can be explained by a basic equation given
below:
$$W(a,b) = \frac{1}{a^n} \int_{-\infty}^{\infty} f(x) \varphi\left(\frac{x-b}{a}\right) dx \qquad (1)$$
Where, f(t) is the time series of our interest;

$\varphi(t)$ is the mother wavelet;

a is the scaling parameter otherwise denoted as the Inverse of Frequency;

b is the Translation parameter, which is directly proportional to Time;

n is the Normalising parameter and equal to 1 generally(say).

The variance of Wavelet coefficients follows power law relation with the scale which can be
given by a simple equation given below;
$$v = x^h$$
Here v is the variance of wavelet coefficients; x is the scale and h is the holder/wavelet
exponent.
Holder/Wavelet exponent provides the measure of roughness/smoothness. If the holder
exponent values are high, it accounts for smoothness whereas low values of holder exponent
emphasis more roughness. After obtaining the holder exponent it can be substituted in the
equation given below to obtain the fractal dimension Value;
$$2D = 5 - h$$
Here, D is the fractal dimension (FD).


**2.3 Discrete Wavelet Transform**
One- dimensional Discrete Wavelet Transform has been carried down in this task as per the
datasets, which are discrete and one dimensional. For the construction of DWT one sets, a =
$2^j$ and b = $2^j$k, where j and k are both integers. 1-D DWT is given by the following equation,
$$D_j(k) = 2^{-\frac{j}{2}} \int_{-\infty}^{\infty} f(t)\, \varphi\big(2^{-j}t - k\big)dt \qquad\qquad (2)$$
Where f(t) is the time series of our interest; K=1, 2, 3...., n, n being the Discrete data array
of maximum Size. Time series of our interest is decomposed to Approximate and Detailed
Coefficients providing both lower and higher frequency information respectively.

**3.0 Results and Discussions**
**3.1 Application to Synthetic data**
A Synthetic signal is generated with three different frequencies such as 3Hz, 5Hz and 10Hz
and analysed by CWT and also applied to synthetic signal added with 25% Gaussian white
noise. The result obtained is shown in Figure 1. As the signal is free from noise possessing
only its own frequencies the mathematical tools didn't posed any difficulty and the
information required are derived without any ambiguity. When the same signal analyzed by
the above mentioned techniques after mixing noise, it provides large difference in the results
which are shown in Figure 2. The CWT provides an acceptable picture in analysing the non-
stationary as well as the same non-stationary signal mixed with noise.CWT not only removes
the ambiguity through by forming wavelet modulus maxima but also through its Wavelet
Coefficients. Also it provides a picture of the Time-Frequency localisation in interpretable
form. An advantage pertaining to wavelet transform is that the Wavelet coefficients records
the exact information of the signal even it is noisy. This notion regarding CWT proves it as a
good tool for identification of lithology in Well logs. Therefore, this technique can be used in
all circumstances to derive the exact information in the Signal.



Mostly, Porosity logs are used for this approach and the fluctuating nature of the porosity
logs can be correlated to both Pores distribution and the fracture (major as well as several
micro fractures) as well. DWT differentiates both fractures and the characteristics of the
pores in the detailed coefficients (Sahimi and Hashemi, 2001). Suppose, the datasets are
collected in a fracture less well than the wavelet detail coefficient (WDC) plot will be
featureless as given below (left of Figure3) but if the datasets are collected in a fractured well
then the WDC plot will be containing highly differentiable features in terms of spikes or local
maxima (right of Figure 3). Same log signal is used in the right one but certain data points are
removed and replaced. The data points which do replaced pertaining to the uniform
distribution constitute both low and high values in comparison with its surrounding data
points. DWT differentiates these particular locations by means of a spike irrespective of the
magnitude of the data points replaced. DWT exposes the signal to low and high frequency
filters produces Detailed and Approximate coefficients respectively.

**3.2 Application of Field Data: Costa Rica Convergent Margin, Central America**
Costa Rica Convergent Margin in Central America is due to the convergence of Cocos and
Caribbean Plates. A seismic migrated section over the region is shown in the Figure
4showing Well sites 1039, 1040 and 1043. Among these wells sites 1039 and 1043 are taken
for study whereas the site 1040 is omitted as it is not passing through certain major litho-
units. Logs such as gamma ray and density are taken for study and the gamma ray signals
exhibiting sharp spikes which are attributed to presence of interbeded ash layers. From the
gamma ray log various lithology are identified and correlated with site adjacent to it. Density
Logs are used for identification of spatial distribution of fractures along the rock matrix using
DWT and WBFA. Core Analysis reports the presence of four sedimentary layers terminated
by a concordant Igneous Intrusion Gabbroic Sill. Well site 1039 is taken as the reference and



lithology identified through Wavelet Transform are correlated to the site-1043 and the result
confirms the subduction zone (Figure 7).

As conventional technique such as Fast Fourier Transform fails in providing the time-

frequency localisation. So, the application of wavelet Transform is the only way to find the
proper time-frequency localisation. The results obtained from CWT analyzed using log data
sets prove the lithological successions. The stratigraphic interfaces occurring in the Well log-
1039 (Figure 5) appears in the Well log-1043 (Figure 6) after having disruptions in the
middle. From the seismic section it is seen that there are four major lithology running from
the Well-1039 to Well-1043 and terminated as Gabbroic Sill. The Well-1040 crossed the
above mentioned strata very mildly and it didn't reach the Concordant intrusive structure as
reached by the Wells-1039 and 1043. Therefore, for interpretation point of view only the
Wells-1039 and 1043 are used. The major successions mentioned after drilling is that the four
sedimentary interfaces followed by a Gabbroic sill. The sedimentary succession obtained
underneath the reference site-1039 situated in the Cocos Plate found to occur in the site-1043
without any disruption. It is also noted from the observation made by Eric et al., (2000) as the
Cocos Plate subducting under the Caribbean Plate the off scarping of the sediments in the
Cocos Plate should occur on the overriding plate but on analysing the chemical composition
it was mentioned the sediment lying on the overriding plate was of different composition.
This analyses comes in support of the effort of framing the subducting system of Costa Rica
using CWT shown in the Figure 7, it is observed that the sedimentary succession in the site-
1039 over the subducting Cocos Plate continuing through the site-1043 without any
disruption situated over the overriding Caribbean plate. In accordance to the locations of the
Wells and the continuity of the sedimentary successions existing in the both sites (1039 and
1043) as traced by the Wavelet scalogram, it is found that the Cocos Plate is subduction
under the Caribbean plate. The lithology identified through time and frequency localisation



tools are used for the WBFA by taking their data points separately. Table 1 shows the FD
values of various lithofacies of both well. From Table 1, we observe that there is transitional
change between sandy and Shaly environments on the basis of variation in FD values and this
variation corresponds to a gradual transition between different sedimentary environments.
Hence, our study suggests that the FD can be used as a well log attribute or even a post-stack
seismic attribute for reservoir characteristic (Brown, 2004).

In Table 1, FD values are greater than 1.2 that emphasises the presence of high shale

content and of low energy environment in depth range between 210 and 330 and between 315
and 430 as reported the presence of sandstone in well site 1039 and 1043 respectively (Figure
10-11). In spite of the presence of sandstone the fractal dimension values are exceeding 1.2
indicating the dominance of shale content and the values are found to be not consistent from
reference site and site 1043. In prior depth ranges, the inconsistency of fractal dimension
values are attributed to the presence of fractures from the structural observations obtained in
well site but in the above mentioned depth ranges the inconsistency as well as from the holder
exponent values it is noted that the roughness exists in the particular lithology. The analysed
results are well correlated with the core samples.

**7. Conclusions**
Lithology identification is a tedious job in well logging and it is the most important one for
reservoir characterisation. To identify Presence of structural feature such as fracture by quick
look interpretation methods is very difficult using well log data. Formation micro imager
(FMI) log often used to identify it is very expensive. Thus methodology used for lithology
and fracture identification using wavelet transform and wavelet based fractal analysis using
holder exponent can be a useful stuff to extract the different lithological feature as well as
stratigraphy feature.



For structural feature identification from various lithologies holder exponent and
fractal dimension values can be utilised and in the presence of some extra information as that
of the structural observations from well sites the results can be more promising. In order to
avoid the assistance of extra information more datasets are needed from the same area so that
on application of WBFA on various lithologies passing through the area provides concrete
idea on lithology and Structural features using holder exponent and fractal dimension values.

**Acknowledgement**
The authors are grateful to editor and Associate editor for critical comments and useful
suggestions to improve our manuscript for publication.

**Figure and Table Captions**
Figure 1: Shows the Continuous Wavelet Transform (CWT) using a synthetic time series

data.

Figure 2: Shows the Continuous Wavelet Transform (CWT) of a synthetic noisy time series

data,

Figure 3: (a) Shows Discrete wavelet Transform (DWT) using synthetic data and original

signal and its DWC-1 below it, (b) synthetic data and original signal edited at certain

points and it's DWC-1 below it.

Figure 4:  Shows the seismic migrated section showing the Wells (after Erik et al, 2000)
Figure 5: showing Continuous Wavelet Transform (CWT) using gamma ray signal and the

Wavelet Coefficient at an altitude-32 of the gamma ray log of the Well location-1039

Figures 6: Showing Continuous Wavelet Transform (CWT) using gamma ray signal and the

Wavelet Coefficient at an altitude-32 of the gamma ray signal of the Well location-

1043




Figure 7: Represents the lithology identification using the gamma ray log of the Well site

1039 and 1043 by the lines drawn on the scalogram and it represents the subduction

zone in the areas obtained from the seismic migrated section.

Figure 8: (a) Shows the discrete detailed and approximate coefficients and the spikes

represents the possible fractures at well location 1039, (b) shows the Discrete detailed

and Approximate coefficients and the spikes represents the possible fractures at well

location 1043

Figure 9: shows the scale of interest shows variance of wavelet coefficients versus scale of

gamma ray of well site 1039 and 1043

Figure 10: Shows variance of Wavelet coefficients versus scale of density log of well site-

1039 and 1043 which shows consistent holder exponent and fractal dimension values

indicating that wells contains similar sedimentary environment.

Figure 11: shows the FD values of both well sites of 1039 and 1043.
Table 1: Shows the FD values of the appropriate lithology identified and the circled depth

ranging and its appropriate fractal dimension values showing deviation of the vales

from the reference site 1039.

Table 2: Shows the ranges of fractal dimension values.

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



Figure 1


























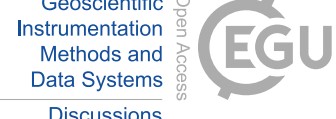



Figure 2

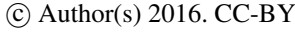





Figure 3(a)




























Figure 3(b)





Figure 4

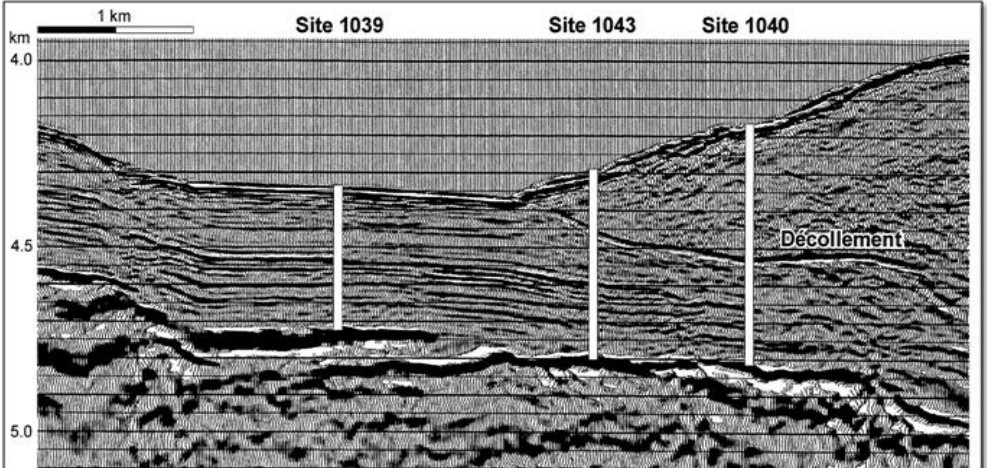





Figure 5

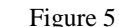





Figure 6




























Figure 7



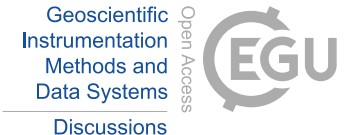

Figure 8

(a)                                                    (b)

499

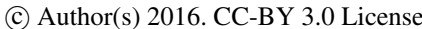



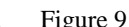

522    Figure 9

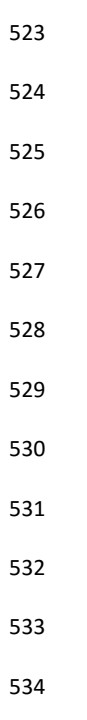
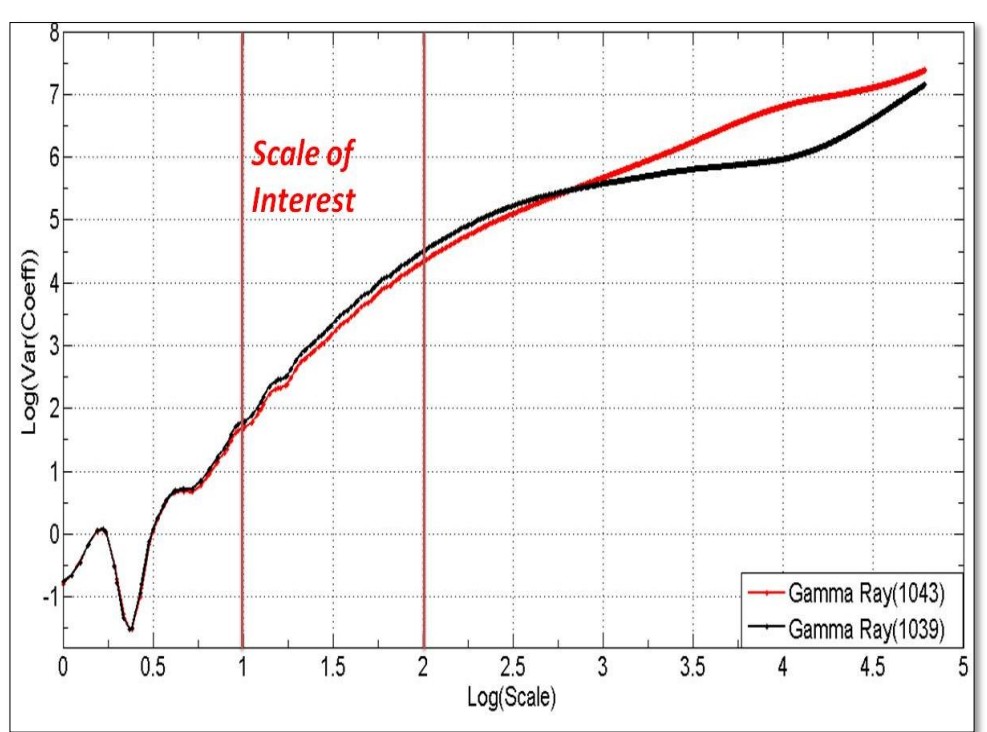





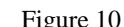

547    Figure 10

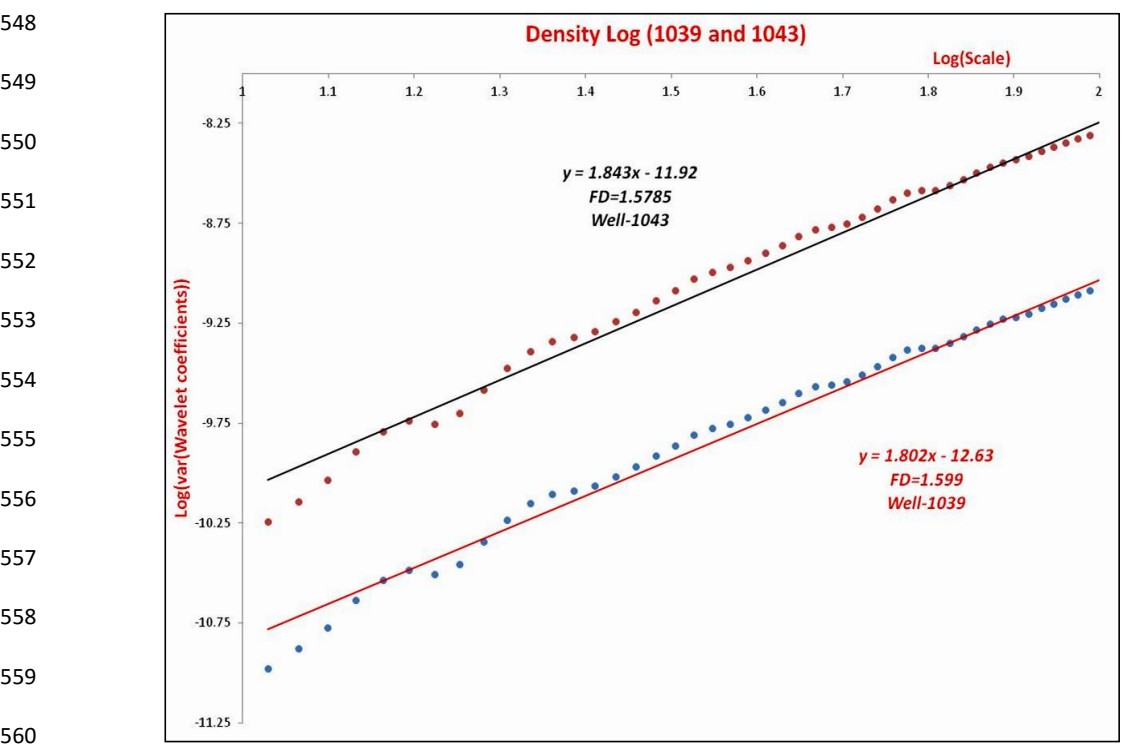



Figure 11

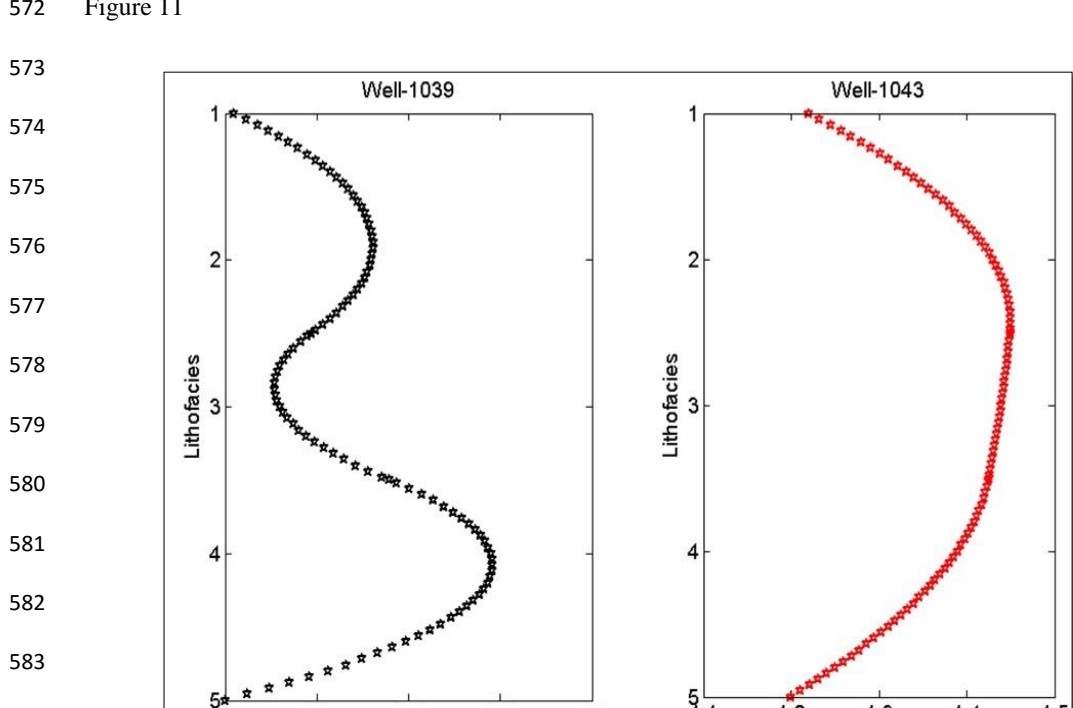




597    Table 1

598

599

| Lithofacies | Depth range(meter) | | Fractal Dimension | |
|---|---|---|---|---|
| | Well-1039 | Well-1043 | Well-1039 | Well-1043 |
| Shale with inter-bedded ash | 20-80 | 60-130 | 1.21 | 1.22 |
| Shaly sandstone | 80-160 | 130-26 | 1.36 | 1.43 |
| Sandy shale with inter-bedded ash | 160-210 | 260-315 | 1.26 | 1.44 |
| sandstone | 210-330 | 315-430 | 1.49 | 1.39 |
| Gabbroic Sill | 330-400 | 430-450 | 1.20 | 1.20 |



Table 2







| Fractal Dimension | Interpretation |
| --- | --- |
| < 0.9 | High sand content and high energy environment |
| 0.9 to 1.2 | Inter-bedded sand and shale |
| > 1.2 | High shale content and low energy environment |