# Peer review of "Geological Stratigraphy and Spatial Distribution of Microfractures over Costa Rica Convergent Margin, Central America- A Wavelet-Fractal Analysis"

_Geoscientific Instrumentation, Methods and Data Systems, 2016_

## Referee Comment (RC1) · Anonymous Referee #1 · 1 Dec 2016

1. Log data is physical measurement based on physical principals which translates log values into direct physical properties of the earth (rocks under observation), for instance Gamma ray log tells about the presence of sand, shale, or any other radioactive element if processed for U, Th logs. How does wavelet transform enhances this understanding about the lithology is not at all clear from the paper. Note that in the logs it is rather easy to identify spikes and other bad log values in combination with the Caliper and bit size log, but once wavelet transform is taken, spikes are translated as discontinuities, and may not necessarily represent real change in lithology.

[Figure]

2. Authors rely on porosity logs for the fracture estimation, again this assumption suffers from the fact that porosity logs are always computed from some logs (density of NPHI) and hence they are not fundamental observations. So any conclusions based on miscalculation in porosity logs will simply percolate to WT. Also, it is in no way clear how WT helps to identify fractures.

3. Please remember that the companies who spend billion dollars for the drilling can always get FMI logs when they need fracture characterization almost for free, compared to drilling cost, so the best contribution from the authors would be to find out fractures away from the wells for instance using seismic data with the help of WT with some good examples.

4. Please correct the typo errors and improve the language.

---

## Referee Comment (RC2) · Anonymous Referee #2 · 13 Dec 2016

The manuscript deals with application of wavelet transform and fractals in identified different lithology and fractures from Gamma ray logs and density logs over Costa Rica Convergent Margin. The study is important for identifying different lithology and fractures from the well logs data. Followings are my major and minor comments:

Major comments:

The continuous wavelet transform highly depends on the choice of mother wavelets. The use of mother wavelet may be discussed and what will be effect of using other mother wavelets in identifying the discontinuities and estimation of fractal dimension
from well logs.

Table 1 changes in fractal dimension values do not seem significant. It will be better to present errors in estimation of holder exponent and fractal dimensions for checking significance of change.

The choice of scale of interest in figure 9 is very subjective and may be discussed in the text. What will be effect of different choices of scale.

Minor Comments:

The term WBFA used in the abstract and other places is not described.

Citation of the figures in the text may be in order. Figure 8 is cited even before figure 1. Figure 7 is cited before figures 5 and 6. Figure 9 is not cited in the text.

Line 104: capital K is not in the equation 2.

Figure 1: Different time scale in top and bottom of figure.

Figure 3 : Deleting few values in figure 3b will be biased and selective. Adding some noise may be better for presenting the results.

The source of well log data may be cited.

Some more key publications on identification of lithology may be cited Fedi (2003) and Bansal et al. (2010) etc.

The Language require major editing.

References: Fedi, M., 2003, Global and Local Multiscale Analysis of Magnetic Susceptibility Data, PAGEOPH, 160,2399-2417.

Bansal, A. R., Gabriel, G. and Dimri, V.P., 2010, Power law distribution of susceptibility and density and its relation to seismic properties: an example from the German Continental Deep Drilling Program (KTB), Journal of Applied Geophysics,72, 123-128.

---

## Author Comment (AC1) · 12 Jan 2017

To

Anonymous Referee manuscript gi 2016-25

Journal of Geoscientific Instrumentation Methods and Data System

Dear Sir First of all we wish a happy and prosperous new year 2017 to you and your family.

We would like to thank to you for suggestions and modifications to improve our

manuscript. Here we have also tried to furnish all those comments given by your side and incorporated in manuscript point to point.

With kind regard Upendra

Reply to First Referee

1. Since Gamma ray log is known for its lithology demarcation capacity and especially this logs are very much noisy in nature, but the lithology variation is indicated as high frequency change in geophysical logs. The concept of Wavelet Transform (WT) is arranging the frequency content present in the signal in descending order once it is plotted with scale (inverse of frequency). In order to validate our interpretation, a wavelet coefficient at a scale above noise level has been extracted and shown along with wavelet scalogram so the non-stationary noises (sudden bursts/sudden ups and downs irrespective of Gamma ray (API) range) which are present are eradicated (Kindly see the wavelet scalograms in Figure 3a and 3b). In reference with core samples the sudden spikes which occur in the gamma ray log is indicated as thin ash beds (Expedition 308 Scientist 2005).

2. As your comment, there a density log is subjected to Discrete Wavelet Transform (DWT), the high frequency (Detailed coefficients) alone extracted and found to produce spikes even the data magnitude is smaller in comparison with its vicinity. For understanding this abnormal behaviour they have taken porosity log for comparison and found to be in good agreement with the detailed coefficient extracted. Kindly see the reference Sahimi, Hashemi, (2001).

3. We have also approached in a similar manner using Wavelet Based Fractal Analysis (WFBA) and Core information. The abrupt changes in fractal dimension and the spike occurring depth range in DWT are found to be correlative and in good agreement with the core information. It is an additional attempt here we made to establish the techniques but it can't be a substitute to any image logs such as Borehole Televiewer or FMI.

Please also note the supplement to this comment:
http://www.geosci-instrum-method-data-syst-discuss.net/gi-2016-25/gi-2016-25-AC1-supplement.pdf
* * *

---

## Author Comment (AC2) · 12 Jan 2017

To

Second Anonymous Referee Journal of Geoscientific Instrumentation Methods and Data System

Dear Sir First of all we wish a happy and prosperous new year 2017 to you and your family.

We would like to thank to you for suggestions and modifications to improve our

manuscript. Here we have also tried to furnish all those comments given by your side and incorporated in manuscript point to point.

With kind regard Upendra

Major Comments: 1. The detail description for selecting the optimum wavelet are mentioned on page 4, line 80-86 in the manuscript. 2. In Table 1, one of the column is included parameter, coefficients of determination (R2) in estimation of holder exponent and fractal dimensions for checking significance of change which is mentioned on page 9, line 197-202 in the text. 3. The choice of scale and its effect in other scale range (Figure 9) were clearly discussed on page 8, line No 163 to 169.

Minor Comments: 1. The detail description of WBFA has been mentioned on page 5. 2. Description of each figure in manuscript are clearly incorporated in order. 3. The term K (Capital) has been replaced by k (small) on page 6 in line 115. 4. In Figure 1, scale of the Figure I is corrected and replaced with new Figure. 5. As your suggestion, the Figure 3 is corrected and replaced with new figure. 6. The source of well log data cited is incorporated and mentioned on page 3, line 51. 7. References has been updated by adding some more publication and some are incorporated accordingly.

Please also note the supplement to this comment:
http://www.geosci-instrum-method-data-syst-discuss.net/gi-2016-25/gi-2016-25-AC2-supplement.pdf

**Supplement:**

[revised manuscript text omitted]

137 fractures) as well. DWT differentiates both fractures and the characteristics of the pores in the 138 detailed coefficients (Sahimi and Hashemi, 2001). For demonstration of the techniques, we have generated two type of synthetic well logs (i) assuming well site is fractureless and (ii) 139 140 well site is fractured. Now wavelet detail coefficient (WDC) for both well site are calculated as sown in in Figure 3(a & b) and Figure 3(c & d). We observed from WDC analysis that will 141 be containing highly differentiable features in terms of spikes or local maxima as shown in 142 Figure 3(d). The noisy data points pertaining to the uniform distribution constitute both low 143 and high values in comparison with its surrounding data points. DWT differentiates these 144 particular locations by means of a spike irrespective of the magnitude of the data points 145 replaced. DWT exposes the signal to low and high frequency filters produces detailed and 146 approximate coefficients respectively. 147

148

**149 **3.2** Application of Field Data: Costa Rica Convergent Margin, Central America**

Costa Rica Convergent Margin in Central America is due to the convergence of Cocos and 150 151 Caribbean Plates. A seismic migrated section over the region is shown in the Figure 4 showing Well sites 1039, 1040 and 1043. Among these wells sites 1039 and 1043 are taken for study 152 153 whereas the site 1040 is omitted as it is not passing through certain major litho-units. Logs such as gamma ray and density are taken for study and the gamma ray signals exhibiting sharp 154 spikes which are attributed to presence of interbeded ash layers. From the gamma ray log 155 156 various lithology are identified and correlated with site adjacent to it. Density Logs are used for identification of spatial distribution of fractures along the rock matrix using DWT. Also 157 the analysis of fractal dimension value through WBFA indicates the presence of fracture in 158 159 lithology. Core Analysis reports the presence of four sedimentary layers terminated by a 160 concordant Igneous Intrusion Gabbroic Sill. Well site 1039 is taken as the reference and 161 lithology identified through Wavelet Transform are correlated to the site-1043 and the result confirms the subduction zone. As conventional technique such as Fast Fourier Transform fails 162 163 in providing the time-frequency localization. So the application of wavelet Transform is the only way to find the proper time-frequency localization. The results obtained from CWT 164 analyzed using log data sets prove the lithological successions. This result is significant in 165 certain scale range only. Since scale is inverse of frequency thus small scale and high scale 166 shows high frequency component and low frequency component of signal. Wavelet analysis 167 of signal at small scale shows the very small changes in data which may be associated with 168 noise also while large scale shows the outspread view of signal. The multiscale analysis has 169 important role in computation of wavelet coefficients (Dimri et al., 2005). The scale is linear 170 171 in a particular range is determined by log(var(cofficients)) versus log(scale) as shown in Figure 5. 172

The stratigraphic interfaces occurring in the Well log-1039 (Figure 6) appears in the Well 173 174 log-1043 (Figure 7) after having disruptions in the middle. From the seismic section it is seen that there are four major lithology running from the Well -1039 to Well - 1043 and terminated 175 176 as Gabbroic Sill. The Well-1040 crossed the above mentioned strata very mildly and it didn't reach the concordant intrusive structure as reached by the Wells-1039 and 1043. Therefore, for 177 interpretation point of view only the Wells-1039 and 1043 are used. The major successions 178 179 mentioned after drilling is that the four sedimentary interfaces followed by a Gabbroic sill. The sedimentary succession obtained underneath the reference site-1039 situated in the Cocos Plate 180 found to occur in the site-1043 without any disruption. It is also noted from the observation 181 182 made by Eric et al., (2000) as the Cocos Plate subducting under the Caribbean Plate the off 183 scarping of the sediments in the Cocos Plate should occur on the overriding plate but on analyzing the chemical composition it was mentioned the sediment lying on the overriding 184 plate was of different composition. This analysis comes in support of the effort of framing the 185 subducting system of Costa Rica using CWT spectrum, it is observed that the sedimentary 186 succession in the site-1039 over the subducted Cocos Plate continuing through the site-1043 187 without any disruption situated over the overriding Caribbean plate. In accordance to the 188 locations of the Wells and the continuity of the sedimentary successions existing in the both 189 sites (1039 and 1043) as traced by correlation of Wavelet scalogram (Figure 8) where 190 191 Figure 8 suggests that the Cocos plate is being subducted under the Caribbean plate. Application of DWT applied to porosity log of both the well 1039 & well 1043 to identify the 192 presence of fracture in lithology figure 9 (a & b). 193

The lithology identified through time and frequency localization tools are used for the 194 WBFA by taking their data points separately. Table 1 shows the FD values of various 195 lithofacies over both wells. The FD values are computed these varies from 1.21 to 1.49 in the 196 197 well-1039 and 1.20 to 1.44 in the well-1043 and associated coefficient of determination,  $R^2$ (%) are also calculated for both wells. We observed the FD and coefficient of determination 198 199 over both wells and found that there are transitional changes between sandy shale and shaly sandstone due to variation in FD values and this variation corresponds to a gradual transition 200 between different sedimentary environments. Hence, the FD values can be used as a well log 201 attribute. 202

Here the FD values are greater than 1.2 that emphasize the presence of high shale content and low energy environment in depth range 210 to 330m and 315 to 430m as reported by (Lopez 2006) in the presence of sandstone over the well sites 1039 and 1043 respectively (Figure 10 &11). In spite of the presence of sandstone, the FD values are exceeding 1.2
indicating the dominance of shale content and these values are found to be not consistent from
reference and 1043 site. In prior depth ranges, the inconsistency of FD values are attributed to
the presence of fractures from the structural observations obtained in well site but in the above
mentioned depth ranges the inconsistency as well as from the holder exponent values it is noted
that the roughness exists in the particular lithology. The analyzed results are well correlated
with the core samples.

213

**214 **7.** Conclusions**

Lithology identification is a tedious job in well logging and it is the most important one for reservoir characterisation. To identify Presence of structural feature such as fracture by quick look interpretation methods is very difficult using well log data. Formation micro imager (FMI) log often used to identify it is very expensive. Thus methodology used for lithology and fracture identification using wavelet transform and wavelet based fractal analysis using holder exponent can be a useful stuff to extract the different lithological feature as well as stratigraphy feature.

For structural feature identification from various lithology holder exponent and fractal dimension values can be utilised and in the presence of some extra information as that of the structural observations from well sites the results can be more promising. In order to avoid the assistance of extra information more datasets are needed from the same area so that on application of WBFA on various lithologies passing through the area provides concrete idea on lithology and Structural features using holder exponent and fractal dimension values.

228

| 229 | Acknowledgement                                                                                     |
|-----|-----------------------------------------------------------------------------------------------------|
| 230 | The authors are thankful to both of the anonymous reviewers for constructing critical               |
| 231 | comments to improve our manuscript.                                                                 |
| 232 |                                                                                                     |
| 233 | Figure and Table Captions                                                                           |
| 234 | Figure 1: Shows the Continuous Wavelet Transform (CWT) using a synthetic time series data.          |
| 235 | Figure 2: Shows the Continuous Wavelet Transform (CWT) of a synthetic noisy time series             |
| 236 | data.                                                                                               |
| 237 | Figure 3: (a) synthetic well logs data over the fractureless well site, (b) Discrete wavelet detail |
| 238 | coefficient (DWC) of fractureless well site, (c) synthetic well log data over the                   |
| 239 | fractured well site, and (d) Discrete wavelet detail coefficient (DWC) of fracture less             |
| 240 | well site.                                                                                          |
| 241 | Figure 4: Shows the seismic migrated section showing the Wells (after Erik et al, 2000)             |
| 242 | Figure 5: shows the scale of interest shows variance of wavelet coefficients versus scale of        |
| 243 | gamma ray of well site 1039 and 1043.                                                               |
| 244 | Figures 6: showing Continuous Wavelet Transform (CWT) using gamma ray signal and the                |
| 245 | Wavelet Coefficient at an altitude-32 of the gamma ray log of the Well location-1039                |
| 246 | Figure 7: showing Continuous Wavelet Transform (CWT) using gamma ray signal and the                 |
| 247 | Wavelet Coefficient at an altitude-32 of the gamma ray log of the Well location-1043                |
| 248 | Figure 8: Represents the lithology identification using the gamma ray log of the Well site 1039     |
| 249 | and 1043 by the lines drawn on the scalogram and it represents the subduction zone in               |
| 250 | the areas obtained from the seismic migrated section.                                               |
|     |                                                                                                     |

.

.

- - -

[revised manuscript text omitted]

341 Figure 1